# Self-Heating Graphene Nanocomposite Bricks: A Case Study in China

**DOI:** 10.3390/ma13030714

**Published:** 2020-02-05

**Authors:** Zhuo Tang, Dong Lu, Jing Gong, Xianming Shi, Jing Zhong

**Affiliations:** 1School of Civil Engineering and Architecture, Wuhan Polytechnic University, Wuhan 430023, China; tangzhuowhpu@163.com; 2Key Lab of Structure Dynamic Behavior and Control (Harbin Institute of Technology), Ministry of Education, Harbin 150090, Heilongjiang, China; dongluhit@163.com; 3School of Civil Engineering, Harbin Institute of Technology, Harbin 150090, China; 4Department of Civil & Environmental Engineering, Washington State University, P.O. Box 642910, Pullman, WA 99164-2910, USA

**Keywords:** nanocomposite bricks, graphene oxide, self-heating, energy consumption

## Abstract

In cold climate regions, the energy associated with indoor heating constitutes a large portion of energy consumption. Increasing energy utilization efficiency is critically important for both economic and environmental reasons. Directly converting electrical energy to thermal energy using joule heating construction elements can save energy and investment to the water pipelines which have been extensively used for indoor heating in China. The fired brick has been extensively used to make pavements, walls and other masonry. Taking advantage of the high dispersion quality of graphene oxide (GO) in water, as well as the firing process used to make fired bricks, graphene nanocomposite bricks with excellent electrical properties and improved mechanical performance were prepared in China. The compressive strength of the bricks showed a substantial increase from 3.15 MPa to 7.21 MPa when GO concentration was 0.1 wt.%. Through applying 5 volts of electrical field within 5 minutes, the nanocomposites can be heated from room temperature to 60 °C, 110 °C and 160 °C for the nanocomposite bricks with graphene concentration of 3 wt.%, 4 wt.% and 5 wt.%, respectively, due to the extremely low percolation threshold (~0.5 wt.%) and high conductivity (10 Ω·cm at 1 wt.%). The sheets were connected more tightly when the GO content was increased. The thermal efficiency can reach up to 88% based on the applied voltage, measured resistance and temperature rise curves.

## 1. Introduction

Indoor heating has been recognized as the main energy consumption source, particularly for cold climate regions—such as North China, that experiences a prolonged period of extreme cold weather (below −20 °C) [1]. It is predicted that the utilization of energy in the building industry will account for more than 35% of the national energy consumption in 2020 in China, of which 25% will be used for heating, according to the report of China Energy Development published in 2018 [2,3]. Currently, in most cold regions in China, the dominant heating method is through heat transfer from hot water, which is essentially provided by coal burning [4]. This not only causes environmental pollution, but also lowers the energy utilization efficiency. This is mainly due to the heat transfers from burning coal to water and then to air which loses energy based on thermal dynamic law. Therefore, searching for new energy resources, such as solar and geothermal energy, as well as increasing the energy usage efficiency—including the employment of a thermal insulation layer with extremely low thermal conductivity—are the two strategies suggested to address the above problems.

In the last decade, a great deal of interest has been focused on phase change materials (PCMs)—which are mainly based on carbonaceous materials like nanotubes (CNTs), and graphene oxide (GO)—to use solar energy to store heat. For instance, Xiang et al. [5] studied the GO nanosheets-modified polyurethane/wood powder composites and obtained a PCM with excellent heat storage properties. Mahdi Maleki et al. [6] used carbon foam to improve different PCMs, and obtained a high light-to-thermal energy conversion efficiency (95%) composite. Although the energy conversion efficiency is improved by PCMs, for indoor heating, the energy generated by solar energy conversion may not be enough. Previous reports indicate that embedding conductive elements in structural members like walls and floors is an effective method to save energy and improve energy efficiency [7]. The 8 μm-diameter conductive steel fibers in cement can achieve a high electrical conductivity of ~1 S/cm through joule heating [8]. It also reported that direct current (DC)-electrical power of 5.6 W (7.1 V, 0.79 A) could raise the temperature to 60 °C [9,10]. However, the high voltage is likely to accelerate the corrosion of steel fibers, and thus the long-term stability of such heating system cannot be guaranteed. In 2009, Chang [11] developed an electric, self-heating concrete system that used embedded carbon nanofiber paper as electric resistance heating elements. The test results showed that a power of 6.4 W (20 V, 0.32 A) could increase the temperature of the mortar from −12 °C to ~10 °C in controlled environment. Unfortunately, due to the limited flexibility of the carbon nanofiber paper, it could only be made as laminate for the application in the mortar cubic, which resulted in very low heat transfer due to the low thermal conductivity of mortar [12,13,14]. Combining conductive materials with construction materials as smaller scale—e.g., nanoscale—is a promising way to simultaneously increasing its joule heating performance and heat transfer dynamics. Kim et al. [15] mixed 2.0 wt.% carbon CNTs with cement to prepare nanocomposites, and the results showed that the temperature increased from 25 °C to 70 °C by applying an external voltage of 10 V. However, the dispersion quality of CNTs is questionable, and the large amount of agglomeration of CNTs can severally deteriorate the mechanical properties of matrix materials [16].

The high dispersion quality of nano additives in matrix is essentially the most important prerequisite for the property enhancement of nanocomposites. The past decades have witnessed the improvement of the dispersion of various nanomaterials in polymer matrix and the corresponding enhanced properties of nanocomposites [17,18]. However, such a scientific advance does not benefit the field of construction materials. The main reason is that it is difficult to modify the intermolecular interaction between nanomaterials and construction materials, like cement and clay which have complex minerals and chemical structures. It has been reported that some admixtures, which are traditionally used in cement industry, can act as dispersant to de-agglomerate nanomaterials with assistant of sonication [19,20]. However, the dispersion quality of nanomaterials is still far from satisfactory, let alone the time and energy consuming during dispersion. Therefore, the search for a nanomaterial, possessing compatible surface properties with construction material and advanced physical properties, is urgent for the application of nano additives.

Graphene, as one type of two dimensional (2D) nanomaterials, which possesses extremely high mechanical, thermal and conductive properties which are all essential characteristics for the application of self-heating [21,22]. In recent decades, graphene has been extensively investigated as nano additive to improve the mechanical and electrical properties of various substrates, including polymer, metal and ceramics due to its high efficiency. Since graphene is very difficult to disperse in almost any solvent with a decent concentration (>10 mg/mL), GO is generally used as graphene derivative [23], it has the geometrical structure of a nanosheet with a thickness less than 1 nm, and the surface is functionalized mainly by -COOH, -OH and -O-, which make the material super-hydrophilic and able to be easily dispersed in water with a very high concentration (up to dozens of mg/mL in a gel state) [21,23]. On the other hand, because of the highly chemical functionalization, GO is insulator with far poorer physical properties than graphene. Therefore, most of the properties of graphene depend on the reduced Graphene (rGO) [23]. Chemical and thermal reduction are currently the two most used methods.

Brick has a long history of being a construction material, and it has been extensively used to make walls, pavements and other elements in masonry construction [24,25]. Among them, fired brick is one of the longest-lasting and strongest building materials, and has been used since circa 4000 Before Christ (B.C.). However, there is still no investigation on graphene-modified clay for its application in bricks. In this study, “Self-Heating Graphene Nanocomposite Bricks: A Case Study in China” was investigated; for the first time, the synthesis of graphene brick nanocomposites with GO was reported. Considering the fabrication of fired bricks involve with high-temperature annealing, it is assumed that this process could transform GO to graphene when it can be embedded in clay matrix before firing. The microstructure, mechanical and self-heating properties were investigated and the test results showed that the compressive strength of the bricks experienced a 28% increase with a dosage of 0.1 wt.% of graphene oxide in the clay. The transformation of GO to graphene was observed and an unprecedented high dispersion of graphene in the clay matrix was achieved, which significantly improved the self-heating performance of the graphene brick nanocomposites.

## 2. Experimental

### 2.1. Materials

Graphene oxide (GO) used in this experiment was synthesized with improved hummer’s method [26], and the concentration of GO solution was 2.85 wt.%. The base of graphene bricks was high white clay, purchased from Zibo, Shandong Province. The chemical composition of high white clay is listed in Table 1. In this study, deionized water was used as the mixing water.

### 2.2. Preparation of Graphene Nanocomposite Bricks

Thirty specimens of cuboid sintered graphene bricks were prepared, the specimens were divided into 10 groups according their GO contents (the reference sample with 0 wt.%, 0.1 wt.%, 0.3 wt.%, 0.5 wt.%, 0.7 wt.%, 1 wt.%, 2wt.%, 3 wt.%, 4 wt.%, and 5 wt.%). Each formulation of graphene bricks had three repetitive samples. The dimension of specimen is 10 mm × 10 mm × 40 mm.

The fabrication process of the graphene bricks in this experiment is presented in Figure 1. Firstly, the high concentration GO solution was diluted with deionized water, and then stirred manually until homogeneous. Then the dried clay powders were weighed and added to the diluted GO solution. Similarly, the mixtures of clay and GO solution (CGOs) were stirred manually to a uniform state without obvious granular substances. Excessive deionized water was added to obtain a homogeneous and suitable flowability of mixture. Secondly, CGOs were put into the oven with the beaker and dried at 50 °C for 12 h until the mixtures were easy to shape. The CGOs were filled in a custom polytetrafluoroethylene (PTFE) mould and dried for 12 h in an oven at the temperature of 50 °C. Brick blanks were then demoulded and dried for another 6 hours in an oven at 80 °C, and then sintered for 6 h in a tube furnace in Argon atmosphere at 900 °C (heating rate was 10 °C/min). Finally, the sintered bricks were polished smoothly on all sides to keep regular shape. The conductive silver paste (produced by Beijing Xingrui Co., Ltd., Beijing, China) was used to fix the fine copper wires on both ends of the brick as the electrodes.

### 2.3. Measurement of Resistivity

During the sintering process, GO was reduced to graphene, which made the graphene bricks have electrical conductivity. Resistivity was employed to evaluate the conductivity of graphene bricks. Low resistance was measured by UNI-T UT51 multimeter and 2450 source test unit (KEITHLEY), and high resistance (more than 200 MΩ) was measured by electrochemical station (Shanghai Chenhua Instrument Co., Ltd, Shanghai, China). In addition, the dimensions of sintered graphene bricks were measured by electronic vernier caliper with the accuracy of 0.01 mm (Guanglu brand). The test process was shown later. The resistivity was calculated according to the following formula:$$\rho =\frac{R\cdot S}{L}$$
where: *ρ* is the resistivity (Ω·cm), *R* is the resistance (Ω), *S* is the cross-sectional area of graphene bricks (cm^2^), and *L* is the length of graphene bricks (cm).

### 2.4. Heating Performance

ATTEN APS3003 Si DC power supply and K-type thermocouple were applied to measure the heating performance of graphene brick. The graphene bricks were placed in a foam box to prevent heat loss, and a thermocouple was attached to the surface of the specimen, then a different voltage was applied, and the changes of temperature were recorded over time.

### 2.5. Mechanical Properties

A specimen with dimensions of 10 mm × 10 mm × 40 mm was used to test flexural strength of graphene bricks. The graphene bricks were cut into three small cube pieces (10 mm × 10 mm × 10 mm) and polished before testing. The WDW-100 computer controlled universal testing machine (Jinan Hengruijin Co., Ltd, Jinan, China) was used to test the mechanical properties of graphene bricks. The loading speed was 0.02 kN/s. The elastic modulus in the compression of graphene bricks was calculated by the slope of the elastic stage of the stress-strain curve. The peak value in each curve was considered the compressive strength of the specimen.

### 2.6. Advanced Characterization of Graphene Nanocomposite Bricks

To investigate the mechanism of graphene bricks and the degree of reduction in GO, X-Ray Diffraction (XRD, 5–90 degrees, XPERT, Malvern Panalytical Ltd., Netherlands) and X-Ray Fluorescence (XRF, Oxford, EBSD) were employed to analyze the phase composition. X-ray photoelectron spectroscopy (XPS, SCALAB 250Xi, Thermo Fisher Scientific Co., Ltd., Shanghai, China) and Fourier Transform Infrared Spectrometer (FTIR Spectrometer, Nicolet IS 50, Thermo Fisher Scientific Co., Ltd., Shanghai, China) were employed to study the changes of the chemical environment of the GO-clay composite after thermal reduction. A scanning electron microscope (SEM, Merlin Compact, Carl Zeiss AG, Jena, Germany) was used to observe the microstructure of specimens.

## 3. Results and Discussion

Figure 1 presents the mixing process of GO-clay suspension. GO gel with various concentration was mixed with clay particles. Because the high dispersion quality of GO in water can be easily guaranteed, we mixed the GO with clay particles by hand mixing for 30 min. The obtained samples became darker with increased GO concentration. To avoid any possible agglomeration of GO in clay material because of the high viscosity of GO gel when the concentration is very high (>~20 mg/mL), clay was mixed with a relatively diluted GO dispersion (10 mg/mL). The water was then evaporated in the oven at 50 °C. In contrast to the white appearance of raw clay powders, the obtained dry powders showed very uniform and smooth yellowish color, indicating that the GO and clay were well-mixed. According to our previous study, GO can wrap clay particles, form core-shell structure, and termed as Clay GO-core shell particles (CG-CSP) [21]. After mixing with a designed amount of water, these CG-CSP swell and gradually turn into gel state, which can be molded into various shapes. Residue water in the sample was removed before firing to prevent crack generation. All the samples were fired at a temperature of 900 °C for 6 h in the atmosphere of Ar.

As can be seen in Figure 1, the higher the graphene concentration, the darker the sample appearance, which can be attributed to the chemical reduction of GO to graphene at a high temperature. Because the graphene-clay brick nanocomposites have high electrical conductivity, this material is termed as conductive graphene clay bricks (CGCB).

The XPS data of specimens in Figure 2 reveals that GO has been mostly reduced to rGO (reduced graphene), because C/O ratio increased from 0.42 to 3.00 after such thermal treatment. There is a large amount of oxygen between the silica tetrahedral and alumina octahedral sheet in the clay, like Si-O and Al-O, and these chemical bonds prevent the oxygen escape from the clay-GO nanocomposite. Besides, the oxygen on the surface of GO, such as –COOH and –OH, is easy to detach under the high temperature thermal treatment [27,28], so the C/O ratio of thermal treated GO may be far beyond 3.00. Compared with the XPS spectra of clay-GO nanocomposite before thermal treatment, the Si/Al-O-C and Si-O-Al bonds in the spectra of Si(2p) and Al(2p) appeared after thermal treatment. This implies that covalent bonding formed between the clay and GO, which is critical for the reinforcement efficiency of graphene [29]. It is also worth noting that the large area of the Si-O-C peak suggests the broad dispersion of GO in the nanocomposite.

The FTIR spectra further describes the changes in the chemical environment of nanocomposite in Figure 3. Before sintering, the characteristic peaks of 3695 cm^−1^, 3618 cm^−1^ and 3445 cm^−1^ reflect the stretching vibration of -OH: 3695 cm^−1^ corresponds to the -OH vibration in the kaolinite interlayer, 3618 cm^−1^ corresponds to the -OH vibration between the kaolinite silicon layer and the aluminum layer, and 3445 cm^−1^ corresponds to the vibration of -OH in the absorbed water in the sample [4,30,31]. Besides, the characteristic peak 1626 cm^−1^ represents the bending vibration of -OH in absorbed water and the vibration of aromatic C=C in GO [29,31]. After sintering, the characteristic peaks at 3695 cm^−1^, 3618 cm^−1^ and 3445 cm^−1^ disappeared due to dehydration and dehydroxylation during heating [4,27,31]. Only a weak peak near 1626 cm^−1^ can be obtained, which is the vibration of aromatic C=C in GO. Before sintering, the peaks in the FTIR spectra corresponding to 1043 cm^−1^, 774 cm^-1^, 691 cm^−1^, and 476 cm^−1^ are generated by Si-O vibration in quartz and kaolinite [29,31]. A peak of 917 cm^−1^ is generated by the bending vibration of Al-OH, and 535 cm^−1^ is generated by the vibration of Al-O in the kaolinite [27]. After sintering, characteristic peaks of metakaolin—1079 cm^−1^,798 cm^−1^ and 471 cm^−1^—are formed, which correspond to the stretching vibration of Si-O, the vibration of Si-O-Al, and the bending vibration of Si-O, indicating the transformation of kaolin to metakaolin [4]. Besides, the occurrence of characteristic peak 560 cm^−1^ implies the formation of γ-Al_2_O_3_ [4,27], which leads to the decrease in compressive strength and activity of calcined kaolinite [4]. In addition, the sintering temperature below 1000 °C can also lower the compressive strength of the fired bricks. Compared with the FTIR spectra tested by Liu et al. [27], it is speculated that mullite and α-cristobalite (alfa-cristobalit) will appear after sintering to 1200 °C, which can greatly increase the compressive strength of specimens.

Figure 4 shows the XRD patterns of the raw material clay and of the clay-GO nanocomposite before and after sintering. It can be seen that the raw material clay mainly contains quartz, muscovite and kaolinite. Similar findings have been reported by Kuang et al. [4], and the corresponded diffraction peak positions are 20.8°, 26.6°, 36.5°, 39.3°, 42.4°, 45.6°, 50.1°, 54.8°, 60° and 68° for quartz, 8.9°, 19.9°, 24.8°, 27.9°, 34.9° for muscovite, and 12.3° and 23.3° for kaolinite. With the addition of GO, the diffraction curve of the clay-GO nanocomposite showed little change except for the weakening of the diffraction intensity. After sintering at 900 °C, the peaks corresponding to muscovite and kaolinite obviously weakened, while the peaks corresponding to quartz had a marginal change. Besides, the diffraction angle of the diffraction peak corresponding to kaolinite at 12.3° increased, indicating that the distance between the layers of kaolinite decreased, which was mainly caused by the removal of intercalated water and of hydroxyl groups between layers [29]. The diffraction peak corresponding to the muscovite at 34.9° disappeared, indicating that part of muscovite structures was destroyed [32].

Figure 5a–c shows the microstructure of CGCB with various GO contents (the reference, 0.7 wt.% and 5 wt.%) at 1000× magnification. From Figure 5a–c, it can be seen that the microstructure of CGCB became denser and more compacted with the increase in GO content. The microstructure of brick specimens at 5000× magnification (Figure 5d–f) show that the minerals, mainly flaky, are distributed evenly on the surface of CGCB. Further, the microstructure of specimens showed that sheets were connected more tightly when GO content increased. Therefore, increasing GO content can improve the integrity of the CGCB, and is beneficial to the formation of a three-dimensional conductive network.

Figure 6 presents the electrical conductivity of samples. It can be found that electrical resistivity dropped from 10^10^ Ω·cm to about 1000 Ω·cm, when the GO content at 0.5 wt.%. Electrical resistivity further dropped to about 100 Ω·cm, when GO content beyond 1 wt.%, which indicating that with the increase in GO content, the electrical resistivity decreased significantly, and the conductivity was significantly optimized. The electrical percolation threshold is dependent on the characteristics of the matrix, nanoadditives and the dispersion quality. For example, Wen et al. [33] found the percolation threshold of 2.6 vol.% in a mixture containing carbon nanotube/carbon black/polypropylene, and obtained a percolation threshold of 0.18 wt.% in the PLA/EVA/rGOs composites. In this study, a percolation threshold of 0.5 wt.% (~0.3 vol.%) was obtained, which is the lowest in all the reported construction-material based nanocomposites (cement, asphalt and other composites) [34]. Such low percolation threshold is mainly due to the high dispersion quality of graphene, as well as the effective reduction method. More specifically, the hydrophilic property of GO allows us to uniformly disperse GO in clay matrix, and the 3D inter-connected GO nanosheets structure can thus be well established. After the annealing process, GO is reduced to graphene with high electrical conductivity.

Figure 7 shows the ion concentration of pure clay solution in terms of zeta-potential. It shows that the amount of ions from clay in the aqueous solution increases as the amount of clay increases, and it achieved a balance about 100 s later. It can be seen that the ions concentration increased to 1.2 × 10^−7^ mol/mL in clay/water with a GO content of 0.2 wt.%, to 7.55 × 10^−7^ mol/mL in clay/water with a GO content of 2 wt.%, and to 4.1 × 10^−6^ mol/mL in clay/water with a GO content of 20 wt.% within 10 min. The dispersion stability of GO is highly dependent on ionic concentration, and a concentration of ~10^−5^ mol/mL could induce the agglomeration of GO [35], which means there is a small time window that skipped us to disperse GO uniformly in clay matrix before the ionic strength becomes too high. When the slats in clay are gradually dissolved in the water, the released metal ions from the clay particles will cross-link the GO nanosheets and physically attract the GO around clay particles. Therefore, GO nanosheets are essentially interlayered between adjacent clay particles. After the thermal reduction process during the sintering process, the rGO (graphene) forms a continuous conductive network.

The self-heating performance of CGCBs were investigated for the potential application in house heating, as presented in Figure 8. It can be seen from Figure 8a that the temperature ramping curve is highly dependent on graphene concentration under a given external voltage. For the samples with 1 wt.% graphene, temperature only increases from 22 °C to 30 °C within 500 s, and then becomes saturated. When the graphene concentration increases to 3 wt.%, the sample can be heated quickly from room temperature to 73 °C in 600 s without any temperature saturation observed afterwards. As the concentration of graphene increases to 5 wt.%, the sample can be heated to 50 °C, 100 °C and 150 °C within 60 s, 120 s and 300 s, respectively. Since all the samples with different graphene concentrations have percolations well above the threshold and the same level of conductivity (from 25.54 to 3.43 Ω·cm), the huge different self-heating behavior of graphene bricks indicated the importance of the density of graphene conductive network embedded in clay matrix. Owing to the low conductivity and high distribution of graphene, the heating rate with low voltage in this study is better than that observed by other researches [8,11], which was about 3.15 °C/300 s in 6.6 W and 33 °C/360 s in 5.6 W.

With the increase in graphene concentration and the electrical conductivity maintained at a stable level, the graphene network became more compacted with more distributed paths for heat generation and transfer, thus sharply enhancing the self-heating properties. Based on the applied voltage, measured resistance, and temperature raising curve, the calculated heating efficiencies were 87%, 88%, 78% for the samples with a graphene concentration of 3 wt.%, 4 wt.% and 5 wt.%, respectively, when the samples were heated from room temperature (RT) to 50 °C, which are much higher than that of coal and gas. This demonstrates that graphene is one of the most conductive electrical/thermal materials. In addition, the self-heating process can be even faster when higher voltage is applied.

Figure 8b illustrates that the sample with 5 wt.% of graphene can be heated from room temperature to 160 °C within 50 seconds. Besides, the temperature raising curves for samples of 4 wt.% and 5 wt.% were almost overlapped with each other, indicating that further increasing graphene concentration to 5 wt.% will not result in a faster heating process or higher final temperature due to the high agglomeration of GO.

The mechanical properties of graphene bricks were tested as shown in Figure 9. The compressive strength of the bricks increased substantially from 3.15 MPa to 7.21 MPa with a GO dosage of 0.1 wt.%, which was followed by a gradual decrease from 5.12 MPa, to the lowest value of 2.41 MPa when the concentration of GO gradually increased from 0.3 wt.% to 3.0 wt.%. This was probably due to the defects and holes of graphene nanosheets and clay matrix, which were induced by gas generation during the reduction process of GO. Afterwards the compressive strength slightly increased to 3.13 MPa and 4.30 MPa with the GO concentration of 4 wt.% and 5 wt.%, respectively. The flexural strength of the bricks also showed an increase from 1.58 MPa to 1.77 MPa when 0.1 wt.% GO was added, and a sharp decrease to 1.38 MPa when the concentration of GO was increased to 0.5 wt.%. The flexural strength then showed a sharp increase to 2.05 MPa as the GO concentration increased to 4.0 wt.%, indicating that a high concentration of GO is beneficial to the increase in flexural strength. The elastic modulus of the reference mix obtained 115 MPa, while the specimen containing 0.1 wt.% GO significantly increased to 243 MPa, which could be attributed to the glue effects of the GO layers; the GO layers can act as a strong 2D adhesive and glue clay particles tightly [21]. The compressive strength, flexural strength, and elastic modulus in Figure 9 showed a similar increasing trend when the concentration of graphene was higher than 3 wt.%, indicating the recovery of the reinforcement effects of graphene. This could be due to the fact that graphene shell encapsulates each clay particle at a high graphene concentration and forms a 3D continuous and strong graphene backbone, which increases the integrity of nanocomposites and thus the strength.

## 4. Conclusions

This study investigated the synthesis of graphene brick nanocomposites with GO, and studied the micro-structure, mechanical and self-heating properties of CGCB. From the experimental results and micro-structure analysis, the following conclusions can be made:(1)High concentration GO is beneficial to the increment of compressive strength, flexural strength and elastic modulus of CGCB. Compressive strength, flexural strength, and elastic modulus showed similar increasing trend when the concentration of graphene was higher than 3 wt.%, indicating the recovery of the reinforcement effects of graphene.(2)There is a small time window that allows us to disperse GO uniformly in clay matrix before the ionic strength becomes too high. With the increase in graphene concentration and the electrical conductivity maintained at a stable level, the graphene network became more compacted with more distributed paths for heat generation and transfer, thus sharply enhancing the self-heating properties.(3)The temperature ramping curve is highly dependent on the graphene concentration under a given external voltage. For the samples with 1 wt.% graphene, the temperature only increases from 22 °C to 30 °C within 500 s, and then becomes saturated. When the graphene concentration increases to 3 wt.%, the sample can be heated quickly from room temperature to 73 °C in 600 s without any temperature saturation observed afterwards.(4)The results from XRD, XPS, FT-IR and SEM proved that graphene was uniformly distributed across the whole matrix, and covalent bonding was formed between clay and GO. The thermal efficiency can reach up to 88% based on the applied voltage, measured resistance and temperature rise curves. The graphene nanocomposite bricks can serve as part of a smart building with using series connection method to share voltage to fit different circumstances, which also sheds lights on the development of intelligent buildings through promoting the combination of nanomaterials and building materials.

## Figures and Tables

**Figure 1 materials-13-00714-f001:**
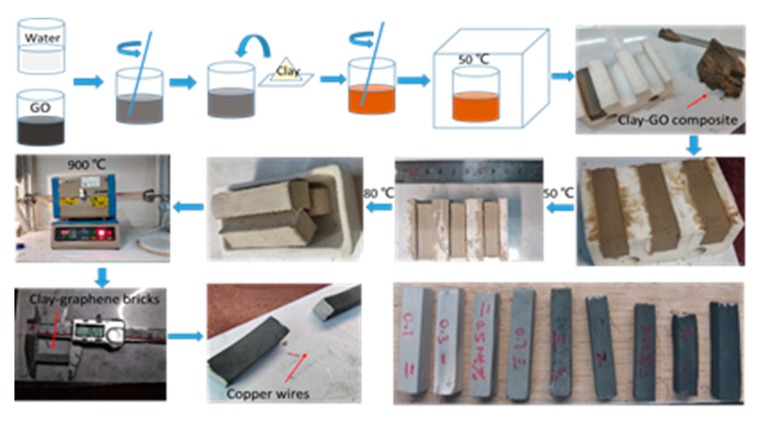
The preparation process of the graphene bricks.

**Figure 2 materials-13-00714-f002:**
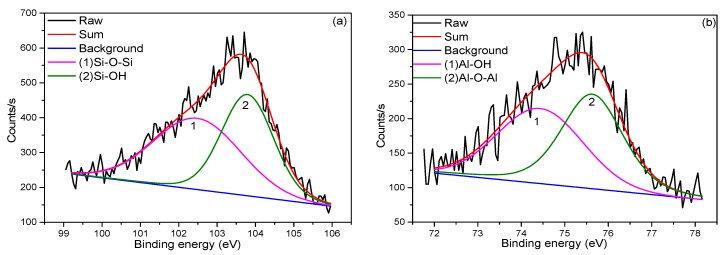
High resolution XPS data for specimens before sintering (**a**) Si2p, (**b**) Al2p, (**c**) O1s, (**d**) C1s and after sintering (**e**) Si2p (**f**) Al2p (**g**) O1s (**h**) C1s.

**Figure 3 materials-13-00714-f003:**
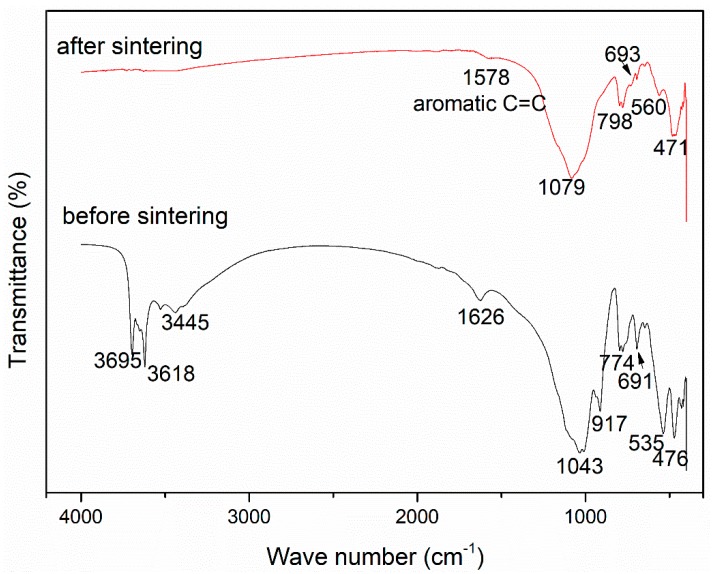
FTIR spectra of graphene bricks before and after sintering.

**Figure 4 materials-13-00714-f004:**
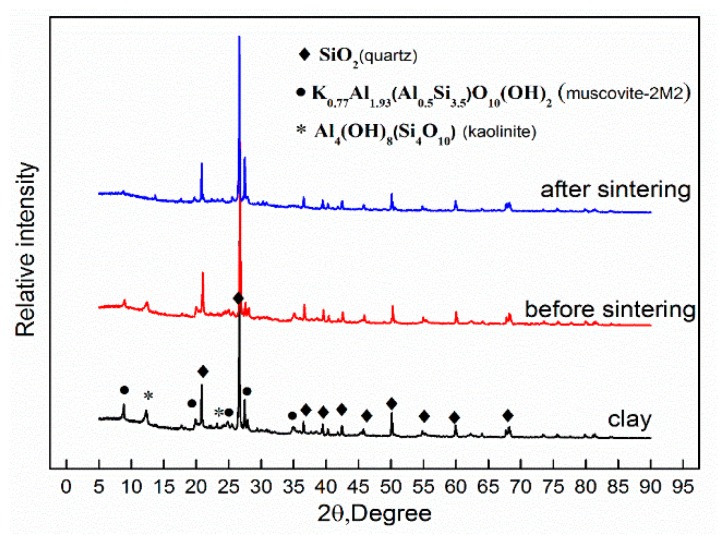
The XRD patterns of clay and graphene bricks before and after sintering.

**Figure 5 materials-13-00714-f005:**
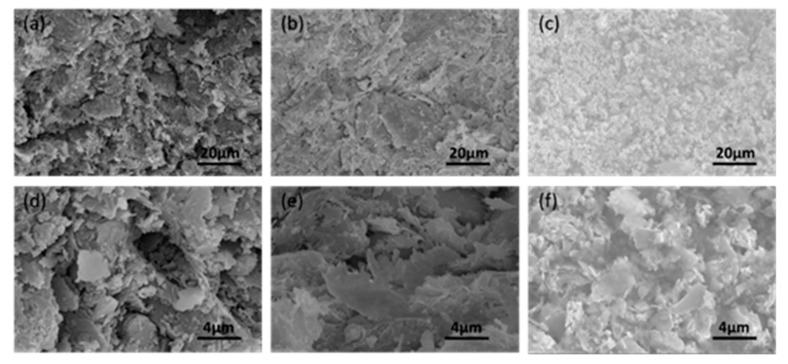
SEM images of clay and conductive graphene clay bricks (CGCB): (**a**) the reference, 1000×; (**b**) 0.7 wt.% GO, 1000×; (**c**) 5 wt.% GO, 1000×; (**d**) the reference, 5000×; (**e**) 0.7 wt.% GO, 5000×; (**f**) 5 wt.% GO, 5000×.

**Figure 6 materials-13-00714-f006:**
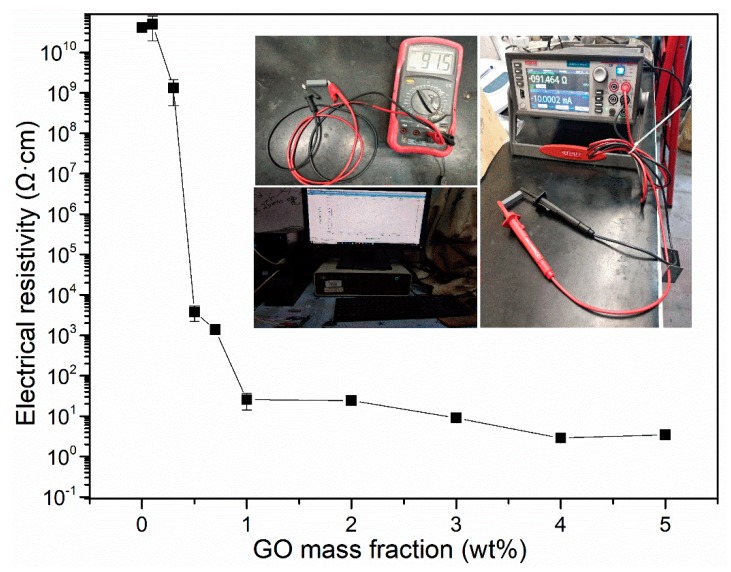
Electrical resistivity of CGCB with different GO contents.

**Figure 7 materials-13-00714-f007:**
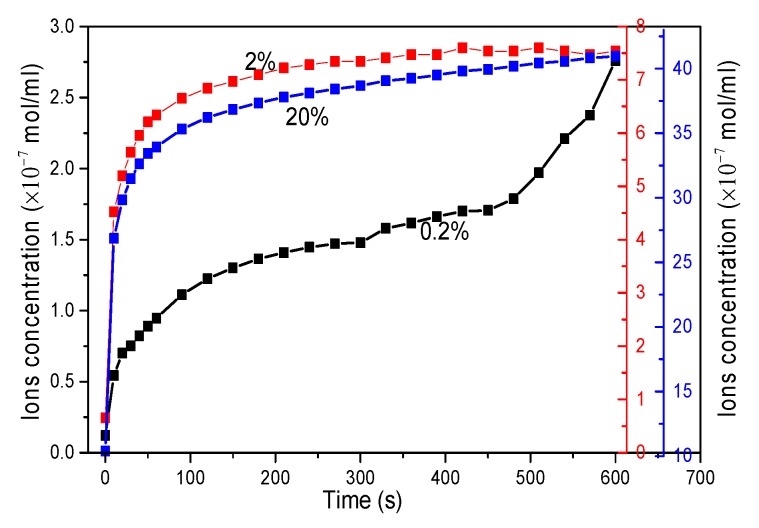
Ions concentration of clay aqueous solutions with different concentrations.

**Figure 8 materials-13-00714-f008:**
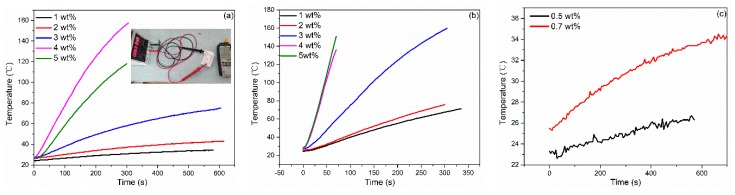
Self-heating performance of CGCB. Heating voltage is 5V (**a**), 10V (**b**), and 30V (**c**).

**Figure 9 materials-13-00714-f009:**
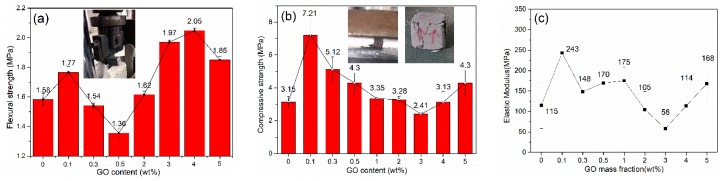
Mechanical properties of CGCB: (**a**) flexural strength, (**b**) compressive strength and (**c**) elastic modulus.

**Table 1 materials-13-00714-t001:** Chemical composition of high white clay (X-ray fluorescence analysis).

Composition (wt.%)	SiO_2_	Al_2_O_3_	K_2_O	Fe_2_O_3_	CaO	Other
Clay	66.9	28.3	3.9	0.4	0.2	0.3

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
