# Peer review of "Self-Heating Graphene Nanocomposite Bricks: A Case Study in China"

_materials, 2020, doi:10.3390/ma13030714_

Round 1
Reviewer 1 Report
The work by Tang et al. is about energy supply in China by the preparation of graphene oxide (GO) dispersion in water and then fabrication of graphene nanocomposite bricks. Modification of clay by the use of such low amount of GO made electrical and mechanical properties improved so that a temperature rise was observed at low loading levels. The idea sounds good, but some major points should be addressed before publication of this work:
1) In the abstract, authors need to specify a parameter that shows effectiveness of GO and the effect of increasing its content. The conductivity and mechanical properties should be taken as response variables and the amount of GO as explanatory variable, then a direct correlation between input and output should be provided by the authors.
2) The methodology and literature survey highlight a case study in China, but authors do not obviously underline whether or not the results are collected or applied in China. If it is a Chinese case study, it should be mentioned in the title by “: A case study in China”, in the abstract and in the introduction and methodology as well.
3) A great deal of interest in the last decade was centered in phase change materials (PCMs), which are not discussed in the introduction. Mainly most of studies are based on carbonaceous materials like CNT, and GO. The following papers should be cited and in the introduction (somewhere suitable) a short paragraph should be dedicated to the use of PCMs, then authors should make some comparisons between their strategy and PCM to highlight the importance and reason for their work:
https://www.sciencedirect.com/science/article/pii/S0927024819307172
https://www.sciencedirect.com/science/article/abs/pii/S0306261919317957
4) In the introduction, authors need to highlight in a better way the importance of their work in comparison with previously published works. This is not clearly specified.
5) Why 2.85 wt% was used as the best concentration in synthesis? Also, GO contents (0 wt%, 0.1 wt%, 0.3 wt%, 0.5 wt%, 0.7 wt%, 1 wt%, 2wt %, 3 113 wt%, 4 wt%, and 5 wt%) were chosen based on which criterion?
6) After XPS data a schematic showing the chemistry of material developed should be provided in the text.
7) SEM micrographs in Fig. 5 should be better explained to understand the differences between them.
8) Electrical threshold should be compared with literature, for some researchers obtained 0.2 wt.%, some other even 0.7 wt.%. The reason should be understood by the reader.
9) Part c of Fig. 9 shows an erratic trend, which should be discussed. There is no trend. Why error bars are so big?
10) In conclusion, authors wrote “The thermal efficiency can reach up to 88% based on the applied voltage, measured resistance and temperature rise curves.” They should aslo metion this in the abstract as the main outcome of their work.
Reviewer 2 Report
The manuscript is dealing with the Self-Heating Graphene Nanocomposite Bricks. This is an interesting topic regarding the nowadays discussions about the self-heating building materials and smart building. The topic is suited for the journal of Materials.
The manuscript reports lot of interesting data and many aspects are discussed in the sections, but there must be major improvements before publishing. Please see the comments in the .pdf document attached and try to correct the paper according the comments.

Reviewer 3 Report
I think the research is very innovative and interesting in a new area of construction Materials.
I suggest in section 2.5, to possibly show photographs of material tests and failure shape of the specimens. For example, compression test and the compressive failure of the samples, etc. This will increase the quality of the paper and would be of the interest of a broader population of researchers.
Round 2
Reviewer 1 Report
The authors considered all comments. Therefore, I suggest "accept".
Author Response
Dear editor and review,
We are so deeply appreciate the time and effort you have spent in our manuscript. Thank you very much.
Best regards,
Jing Zhong
Reviewer 2 Report
Dear authors,
you corrected the manuscript according my comments, and now the manuscript is presenting very interesting research and evaluation of the results in much better way. I have only few comments now, before the paper will be published, please check and try to correct the following details:
P2L55 - DC-electrical - please explain abbreviation "DC" firstly, then use only short
Table 1 - I suggest to put into brackets only "(X-ray fluorescence analysis)
P4L118 - had instead of "has"
P4L127 - Polytetrafluoroethylene is written together grammatically correct
P4L124 - please correct the sentence "Excessive mixing water was added into the mixing material.." to make it correct
P4 - When you use "CGO were.." gramatically correct is "CGOs were" - plural
P5L172 - Use "On the contrary,...."
P5L177 - "All the samples were fired"
P5L181 - "...the Gr/clay..." - do you mean graphen/clay ?
P7L212 - i think that "a-cristobalite" should be "α-cristobalite" (alfa-cristobalit)
PL239 - "...electrical conductivity had significantly optimized..." - what do you mean by this expression ? What electrical conductivity optimized ? It should be " Conductivity wassignificantly optimized" ? - please correct
P10 - Discussion about the results of elastic modulus is missing - please add, two sentences will be enough. And please specify a elastic modulus you tested - is it elastic modulus in compression or ? The value of 243 MPa is very high for brick - please explain this high value
After the correction of above comments and English grammar corrections, I recommend the paper to be published.
